# Zero-shot Clustering of Embeddings with Pre-trained and Self-Supervised Learning Encoders

## Abstract

In this work, we explore whether pretrained models can provide a useful representation space for datasets they were not trained on, and whether these representations can be used to group novel unlabelled data into meaningful clusters. To this end, we conduct experiments using image representation encoders pretrained on ImageNet using either supervised or self-supervised training techniques. These encoders are deployed on image datasets that were not seen during training, and we investigate whether their embeddings can be clustered with conventional clustering algorithms. We find that it is possible to create well-defined clusters using self-supervised feature encoders, especially when using the agglomerative clustering method, and that it is possible to do so even for very fine-grained datasets such as iNaturalist. We also find indications that the Silhouette score is a good proxy of cluster quality for self-supervised feature encoders when no ground truth is available.

## 1 Introduction

Self-supervised learning (SSL) has seen a large amount of interest in recent years across almost every machine learning sub-field, due to the promise of being able to harness the large quantities of unlabelled data available and obtaining generic feature embeddings useful for a variety of downstream tasks Balestriero et al. (2023). This has for example led to the development of impressive large language models (Brown et al., 2020) and computer vision systems trained on 1 billion images Goyal et al. (2021).

However, while the embeddings from an SSL-trained feature extractor can perform well on downstream tasks after fine-tuning the network, there has been little investigation into the utility of the embeddings without fine-tuning the network. Prior work by Vaze et al. (2022) and Zhou & Zhang (2022) suggests that the SSL feature encoders generate embeddings that are suitable for clustering, but nonetheless still further adjust the feature encoders through fine-tuning. Yet, widespread interest in application of large pre-trained models on custom datasets, combined with prohibitive cost of compute, make this question important and increasingly urgent

We find that there has so far been no investigation into whether SSL-trained feature encoders can generate informative clusters of embeddings on datasets that were totally unseen to the encoder. In this work, we therefore perform a zero-shot transfer learning task, evaluating the performance of a suite of SSL-trained feature encoders across a diverse set of datasets, using different classical clustering methods. In summary, we make the following contributions:

- We conduct the first investigation into zero-shot clustering of SSL feature encoders, finding that Contrastive and Multi-Modal SSL approaches can produce meaningful clusters across a variety of datasets without per-dataset parameter tuning.
- We find that the Agglomerative Clustering method is consistently strong across SSL encoders, backbones, and datasets.
- We find that the Silhouette score is highly correlated with the Adjusted Mutual Information score, and can be a strong proxy of clustering performance without access to ground truth labels.

## 2 RELATED WORK

Our work builds upon two broad fields of research: self-supervised learning for computer vision applications, and clustering. We give a general overview of each field.

**Self-Supervised Learning** (SSL) has recently received an increasing amount of interest from the computer vision domain, in part due to its promising results in natural language processing (Brown et al., 2020). Whilst SSL has a long history of research, the currently dominant methods can be divided into five general categories as follows (Balestriero et al., 2023). (1) Contrastive Learning approaches, which build on metric learning, in which embeddings of multiple views of the same instance are brought together and embeddings from different instances are pushed apart Chen et al. (2020); He et al. (2020); Chen et al. (2021). (2) Self-Distillation approaches, where a student and teacher encoder process an input image with distinct transforms applied, and the student is tasked with predicting the embeddings of the teacher (Grill et al., 2020; Chen & He, 2021; Caron et al., 2021; Oquab et al., 2023). (3) Canonical Correlation Analysis approaches, where the feature embeddings are analyzed in terms of the cross-covariance matrix, through mechanisms such as minimizing covariance across feature dimensions and minimizing correlation across feature embeddings for different inputs (Zbontar et al., 2021; Bardes et al., 2022). (4) Masked Image Modelling approaches, where large parts of the input image are masked out and have to be reconstructed in image-space (He et al., 2022; Zhou et al., 2022; Bao et al., 2022). (5) Multi-Modal Learning approaches, where the utilized data consists of different modalities, such as image-text pairs, which are separately embedded and must be aligned (Radford et al., 2021).

**Clustering** is one of the most common tasks in a large variety of applications and can be defined as the task of finding local structures that are homogeneous and separated without explicit label supervision (Everitt et al., 2011). This problem has been studied for centuries resulting in methods using clustering criteria based on partitioning (Lloyd, 1982; Arthur & Vassilvitskii, 2007), fuzzy theory (Bezdek et al., 1984), graph theory (Frey & Dueck, 2007; Yu & Shi, 2003), density (Ankerst et al., 1999; Ester et al., 1996; McInnes & Healy, 2017), hierarchies (Ward, 1963; Sokal & Michener, 1958), and many more (Xu & Tian, 2015). These methods have traditionally necessitated a disjointed processing pipeline, as the clustering algorithms have been optimized independently of the feature generators. However, in recent years several methods have been proposed to jointly learn feature extractors and clustering processes (Pakman et al., 2020; Caron et al., 2018; Tapaswi et al., 2019; Ronen et al., 2022; Yang et al., 2017; Van Gansbeke et al., 2020).

## 3 EXPERIMENTAL DESIGN

We consider the task of **zero-shot clustering** of feature embeddings obtained from pretrained self-supervised encoders. The aim of this task is to cluster the feature embeddings from various as-yet unseen datasets, in a way such that the clusters are intrinsically well-defined and, ideally, match the ground truth label assignments. Our feature encoders and clustering methods are only tuned on data from a single dataset, the commonly used ImageNet-1k (Russakovsky et al., 2015). This methodology is then deployed on all other tested datasets without re-tuning any of the parameters.

### 3.1 FEATURE ENCODERS

In order to capture the diverse methodologies within the self-supervised learning field, we compare methods from the major self-supervised paradigms within computer vision (Balestriero et al., 2023). We choose one representative method per paradigm, and compare the clusterability of their features against those of a model pretrained with cross-entropy supervision using the ImageNet-1k labels. The SSL models selected are as follows:

- **Contrastive Learning**: MoCo-v3 (Chen et al., 2021)
- **Self-Distillation**: DINO (Caron et al., 2021)
- **Canonical Correlation Analysis**: VICReg (Bardes et al., 2022)
- **Masked Image Modelling**: MAE (He et al., 2022)
- **Multi-Modal Learning**: CLIP (Radford et al., 2021)

For each method we consider two common backbone networks, ResNet-50 (He et al., 2016) and ViT-B (Dosovitskiy et al., 2021) trained on the ImageNet-1k dataset, using publicly available checkpoints. However, it should be noted that (1) the MAE model only supports transformer architectures and so does not have a ResNet-50 checkpoint; (2) VICReg does not have a pretrained ViT-B checkpoint; and (3) the CLIP model makes several modifications to the backbone architectures. Furthermore, the CLIP model was not trained on ImageNet-1k, and was instead trained on a different, non-disclosed, large dataset of paired images and text captions. We include the CLIP model nonetheless since it has previously been shown to perform well on zero-shot classification tasks when supplied with text embeddings of the classes against which to compare (Radford et al., 2021).

## 3.2 CLUSTERING METHODS

In order to cluster the feature embeddings, we considered several classical clustering methods: K-Means (Lloyd, 1982) (with K-Means++ initialization Arthur & Vassilvitskii (2007)), Agglomerative Clustering (AC) (Everitt et al., 2011), Affinity Propagation (Frey & Dueck, 2007), and HDBSCAN (McInnes & Healy, 2017). These clustering methods were chosen because they have few hyperparameters to tune, cover several clustering paradigms (partition, hierarchical, graph-theory, and density), and include both parametric and non-parametric methods. As K-Means requires the number of clusters in order to run, we assume that this is known *a priori*. In contrast, HDBSCAN, AC, and Affinity Propagation automatically determine the number of clusters in the data, with AC also optionally able to operate with the number of clusters defined beforehand.

Among these, HDBSCAN can identify samples which belong to *no* cluster (noise samples). Unless stated otherwise, we consider the noise class to be its own class when computing the AMI (see Eq. 2). This unfortunately sets HDBSCAN at a disadvantage, since the samples it identifies as noise are typically distributed across all ground-truth classes, but is fairer than ignoring samples it identifies as noise since that would evaluate it only on easier samples.

We actively choose not to consider neural clustering methods, such as Neural Clustering Processes (Pakman et al., 2020) or DeepCluster (Caron et al., 2018), as these methods jointly learn the feature encoder and clustering step, which is outside the scope of our investigation. In this work, we focus solely on how well the feature embeddings of pretrained self-supervised encoders can be clustered.

## 3.3 DATASETS

We evaluate the different permutations of feature encoders and clustering methods on a diverse set of datasets, see Table 1. These datasets span tasks with differing levels of label granularity, number of classes and samples, domain shifts, and degree of class imbalance. Out of all these datasets only the ImageNet training split has previously been observed during training of the feature encoders[1] as well as setting the hyperparameters of the clustering method. All other datasets have not previously been observed and the considered methods are not tuned in any way on these.

## 3.4 EVALUATION METRICS

We evaluate the performance of a clustering using two metrics: Adjusted Mutual Information (AMI) (Vinh et al., 2009) and the Silhouette score (Rousseeuw, 1987). AMI measures the agreement between the constructed clusters and the ground truth clustering, while the Silhouette score measures how well-defined the clusters are irrespective of whether the cluster elements are correctly assigned.

### 3.4.1 ADJUSTED MUTUAL INFORMATION

Since we are evaluating the clustering on annotated datasets, we evaluated a candidate clustering assignment against the "ground truth" cluster labels, from an information theoretic perspective. The Normalized Mutual Information (NMI) between two label assignments $V$ an $U$ is defined as

$$\text{NMI}(U, V) = \frac{\text{MI}(U, V)}{\text{mean}(\text{H}(U) + \text{H}(V))}, \tag{1}$$

---

[1]Except potentially the CLIP models, for which we don't know whether or not it was trained on these datasets.

Table 1: **Dataset overview.** For our zero-shot clustering protocol we consider a diverse set of experiments of differing levels of task granularity, number of classes and samples, domain shift, and class imbalance. The reported numbers are on the publicly available test splits. If the test labels are not publicly available the public validation split is used instead. The class imbalance, $\rho$, is measured with the ratio between the number of samples in the largest and smallest classes in the dataset.

| Dataset | #Samples | #Classes | $\rho$ | Description |
|---|---|---|---|---|
| ImageNet-1k (Russakovsky et al., 2015) | 50,000 | 1,000 | 1.00 | Diverse general objects |
| CIFAR10 (Krizhevsky, 2009) | 10,000 | 10 | 1.00 | Diverse general objects |
| CIFAR100 (Krizhevsky, 2009) | 10,000 | 100 | 1.00 | Diverse general objects |
| MNIST (Lecun et al., 1998) | 10,000 | 10 | 1.27 | Handwritten digits |
| Fashion MNIST (Xiao et al., 2017) | 10,000 | 10 | 1.00 | Clothing articles |
| SVHN (Netzer et al., 2011) | 26,032 | 10 | 3.20 | House numbers |
| Oxford Flowers (Nilsback & Zisserman, 2008) | 6,149 | 102 | 11.90 | Flower variants |
| FGVC Aircraft (Maji et al., 2013) | 3,333 | 100 | 1.03 | Aircraft variants |
| NABirds (Van Horn et al., 2015) | 24,633 | 555 | 6.67 | Bird species |
| iNaturalist (2021) (Van Horn et al., 2021) | 100,000 | 10,000 | 1.00 | Plant & animal species |

where $\text{MI}(U, V)$ is the mutual information between label assignments $V$ an $U$, and $H(\dot{)}$ is the Shannon entropy of the considered label assignment. NMI is a relative measure of the amount of information between two label sets, and hence is bounded between 0 and 1 with 1 occurring for a perfect match, and 0 occurring when there is absolutely no mutual information between the label assignments.

However, NMI is not corrected for chance so its value can increase merely by increasing the number of clusters used (Vinh et al., 2009). In order to account for this, we use the Adjusted Mutual Information metric proposed by Vinh et al. (2009), defined as

$$\text{AMI}(U, V) = \frac{\text{MI}(U, V) - \mathbb{E}[\text{MI}(U, V)]}{\text{mean}(\text{H}(U) + \text{H}(V)) - \mathbb{E}[\text{MI}(U, V)]}, \tag{2}$$

where $\mathbb{E}[\text{MI}(U, V)]$ is the expected value of the mutual information between the considered label assignments. Similar to NMI, an AMI of 1 represents a perfect agreement between label assignments, but a score of 0 indicates the typical score for a completely random label assignment (negative AMI scores are possible).

### 3.4.2 SILHOUETTE SCORE

The Silhouette score, $S$, is a clustering measure based on the intrinsic structure of the created clusters (Rousseeuw, 1987), defined as

$$S = \frac{1}{N} \sum_{i}^{N} \frac{a_i - b_i}{\max(a_i, b_i)}, \tag{3}$$

where $N$ is the total number of data points, $a_i$ is the average distance between data point $i$ and all other points assigned in the same cluster, and $b_i$ is the average distance from $i$ to all points in the next nearest cluster. $S$ is bounded between $-1$ and $1$. A score near 0 indicates that clusters are overlapping, as the data points are equally close to several clusters. A score of 1 indicates that the clusters are dense with little within-cluster distance, and thereby well-clustered. Negative values may indicate an inaccurate clustering. Since $S$ is defined based on the relative distances of data points, it can be computed without reference to a set of ground-truth cluster assignments.

### 3.5 HYPERPARAMETER SEARCH

In order to maximize the performance of each permutation of the feature encoder and clustering methods, we conducted a staggered sweep over the relevant clustering hyperparameters. The sweep was conducted using subsets of the training splits of ImageNet-1k, Imagenette, and Imagewoof (Howard). Imagenette and Imagewoof are coarse- and fine-grained subsets of ImageNet-1k, respectively, with 10 classes each. These datasets were selected to find hyperparameters which were robust against changing the number of classes and their granularity, whilst only optimizing clustering performance on data within the encoder's original training set.

For each of the three datasets, we created a validation set by taking a class-stratified random subset of the training set, using the same number of samples as appeared in the datasets' test set (50000, 3925, and 3929 respectively). The same split was used across all encoders, clusterers, and stages of the hyperparameter search. For Affinity Propagation, it was not feasible to conduct this search on ImageNet due to compute and memory scaling w.r.t. number of samples; hence we optimized Affinity Propagation hyperparameters using Imagenette and Imagewoof only.

First, as the curse of dimensionality can negatively affect the performance of the considered clustering methods (Bellman, 1966), we searched for an appropriate dimensionality reduction process. We compared the performance of using the original un-reduced feature embedding space (up to 2048-d) against applying PCA (Pearson, 1901) or UMAP (McInnes et al., 2018) to reduce the number of dimensions. Specifically, we considered reducing the feature embeddings to $[2, 5, 10, 20, 50, 100, 200, 500]$ with either PCA or UMAP, and considered reducing the number of dimensions to capture a target fraction of total variance of the data $[0.75, 0.8, 0.85, 0.9, 0.95, 0.98, 0.99]$. To perform PCA, we first took the z-score of each dimension and then used the default hyperparameters of SCIKIT-LEARN (Pedregosa et al., 2011), without whitening the data. To perform UMAP, we increase the number of neighbours considered to 30 and decreased the minimum distance to 0, following the recommendations of (McInnes, 2018); we otherwise used the default hyperparameters of UMAP (McInnes et al., 2018). In this first stage, we used the default hyperparameters of the clustering methods as defined in SCIKIT-LEARN. For K-Means and AC, we provided the number of annotated classes within the dataset (1000 or 10) as number of clusters to produce. For each encoder and clusterer, we took the average AMI over the three datasets and selected the method which yielded the highest average AMI (a particular PCA dim, PCA variance, UMAP dim, or no reduction).

We observed that for K-Means, AC, and HDBSCAN, the majority of encoders all performed best with UMAP-reduced embeddings and were insensitive to the choice of dimension, with minimal change in mean AMI across the range 5 to 500. Thus for consistency, we selected a 50-dim UMAP reduction for all encoders/clusterers where UMAP performed best. The MAE-trained ViT-B encoder bucked this trend and performed poorly with UMAP reduction across all clusterers (and all three datasets). For Affinity Propagation, PCA outperformed UMAP (as it failed to converge on UMAP-reduced embeddings); most encoders worked best with a 10-dim PCA reduction.

In the second stage, using the dimensionality reductions per encoder from the first stage, we iterated over the per-method specific hyperparameters for AC. Continuing to use the number of classes as the number of clusters, we evaluated all combinations of distance metric ($\ell_1$, $\ell_2$, $\ell_\infty$, cosine) and linkage method (ward [$\ell_2$ only], complete, average, single), for 13 options in total. For each encoder, we selected the metric and linkage which yielded the best average AMI over the three datasets. The selected options were $\ell_2$ + ward (5 encoders), $\ell_2$ + avg (3 encoders), or $\ell_\infty$ + avg (2 encoders).

Thirdly, we tuned the distance threshold to use for each encoder. The distance threshold provides an alternative stopping criteria for AC so it does not need to know the number of clusters *a priori*. For each encoder, we fit the clusterer on each of the 3 datasets for 21 distance thresholds sampled logarithmically from 0.001 to 5000.0, and then selected the distance threshold which yielded the highest average AMI.

For HDBSCAN, we noticed that for some encoders it would select very few clusters for Imagenette and Imagewoof, reducing its performance. We verified, by clustering the full embeddings, that decreasing the maximum cluster size mitigated this problem. We thus set the maximum cluster size to be a generous 20% of the number of samples throughout the remainder of the experiments, so as to ensure HDBSCAN produced more than a couple of clusters but without constraining it too much.

## 4 EXPERIMENTAL RESULTS

We report the zero-shot clustering capabilities of the considered SSL feature encoders and clustering methods measured by AMI in Table 2a and Table 2b, for ResNet-50 and ViT-B backbones, respectively.

Across both the ResNet-50 and ViT-B backbones, the best performance on ImageNet-1k (the dataset used for training) and CIFAR-10 and CIFAR-100 (the datasets most similar in their domain to ImageNet) is obtained using the encoders trained with conventional classification supervision. For Im-

Table 2: **AMI scores of SSL encoders and clustering methods.** We report the AMI score, as a percentage, on each test dataset (see Table 1) for each encoder and clusterer. The performance of agglomerative clustering is shown twice: once using the ground-truth num. of classes as the num. of clusters (AC w/ C), once predicting the num. of clusters (AC w/o C). The hyperparams of the clustering methods are only tuned on ImageNet-1k (IN1k). The best combination of encoder/clusterer per dataset and backbone **bolded**; the best encoder per clustering method is underlined. *Some Affinity Propagation results could not be obtained due poor memory and compute scaling with $n$ samples.*

(a) **AMI score (%) with a ResNet-50 backbone.**

| | Encoder | IN1k | C10 | C100 | MNIST | fMNIST | SVHN | Flowers | Aircraft | NABirds | iNat21 |
|---|---|---|---|---|---|---|---|---|---|---|---|
| K-Means | Supervised | **73.0** | **68.2** | **51.5** | 81.3 | 68.7 | 4.9 | 63.9 | 14.6 | 38.7 | 8.7 |
| | MoCo-v3 | 48.4 | 63.9 | 50.9 | **86.5** | **71.2** | 11.2 | 80.3 | 20.7 | 28.0 | 4.4 |
| | VICReg | 45.8 | 52.7 | 44.8 | 79.8 | 69.9 | 3.0 | 81.2 | 16.1 | 18.4 | 3.8 |
| | DINO | 44.3 | 49.3 | 41.5 | 74.1 | 64.2 | 0.9 | 81.5 | 17.7 | 18.1 | 3.6 |
| | CLIP | 50.3 | 48.5 | 39.9 | 54.3 | 52.7 | 1.4 | 82.5 | 29.9 | 42.2 | 8.2 |
| AC w/ C | Supervised | **73.0** | 67.3 | **51.8** | 81.6 | 68.9 | 3.7 | 64.1 | 15.2 | 39.0 | 8.9 |
| | MoCo-v3 | 48.7 | 63.5 | 50.6 | **86.7** | 70.2 | 10.1 | 81.0 | 20.4 | 28.3 | 4.5 |
| | VICReg | 46.1 | 53.0 | 45.2 | 79.2 | 69.3 | 1.4 | 81.8 | 16.4 | 19.0 | 3.9 |
| | DINO | 48.0 | 48.0 | 41.6 | 74.0 | 66.9 | 0.7 | 82.2 | 19.5 | 21.4 | 6.7 |
| | CLIP | 50.1 | 52.2 | 38.8 | 81.2 | 61.1 | 1.1 | **85.9** | 31.4 | 44.3 | 9.8 |
| AC w/o C | Supervised | 63.6 | 67.3 | 49.5 | 74.2 | 65.9 | 5.9 | 56.9 | 17.4 | **47.6** | 22.0 |
| | MoCo-v3 | 48.3 | 64.0 | 46.0 | 82.4 | 68.2 | **13.0** | 70.5 | 16.9 | 32.5 | 14.9 |
| | VICReg | 47.1 | 52.6 | 43.2 | 76.0 | 66.5 | 4.6 | 72.0 | 10.3 | 25.9 | 14.0 |
| | DINO | 46.9 | 46.5 | 39.8 | 70.4 | 62.8 | 2.9 | 79.0 | 17.2 | 25.4 | 15.9 |
| | CLIP | 49.7 | 48.7 | 38.7 | 75.8 | 55.9 | 1.6 | 83.1 | **33.2** | 43.9 | **23.2** |
| Affinity Prop | Supervised | - | 40.2 | 39.4 | 44.3 | 42.4 | 9.6 | 52.8 | 12.2 | 37.0 | - |
| | MoCo-v3 | - | 38.1 | 29.7 | 46.4 | 45.2 | 11.9 | 46.4 | 14.6 | 20.0 | - |
| | VICReg | 12.5 | 31.9 | 25.1 | 39.0 | 43.0 | 6.5 | 48.8 | 12.8 | 17.9 | - |
| | DINO | - | 31.1 | 30.4 | 43.0 | 40.5 | 7.9 | 59.9 | 16.8 | 18.1 | - |
| | CLIP | - | 35.8 | 22.2 | 44.3 | 41.1 | 5.8 | 61.1 | 24.8 | 28.2 | - |
| HDBSCAN | Supervised | 64.0 | 36.6 | 42.7 | 70.3 | 48.6 | 5.6 | 55.9 | 10.3 | 27.8 | 7.8 |
| | MoCo-v3 | 34.0 | 36.6 | 38.7 | 76.9 | 45.5 | 10.6 | 76.3 | 14.3 | 25.6 | 5.1 |
| | VICReg | 32.8 | 29.7 | 33.0 | 72.6 | 48.7 | 5.3 | 77.0 | 11.6 | 14.3 | 4.9 |
| | DINO | 29.5 | 27.5 | 27.0 | 68.2 | 44.5 | 3.4 | 77.3 | 13.2 | 18.3 | 3.8 |
| | CLIP | 31.1 | 25.2 | 20.6 | 77.9 | 40.6 | 2.9 | 73.9 | 28.0 | 29.2 | 10.3 |

(b) **AMI score (%) with a ViT-B backbone.**

| | Encoder | IN1k | C10 | C100 | MNIST | fMNIST | SVHN | Flowers | Aircraft | NABirds | iNat21 |
|---|---|---|---|---|---|---|---|---|---|---|---|
| K-Means | Supervised | **78.5** | **82.6** | 65.2 | 80.5 | 69.7 | 1.3 | 68.0 | 18.3 | 37.8 | 8.4 |
| | MoCo-v3 | 59.9 | 78.8 | 62.1 | 82.6 | 71.0 | 1.4 | 80.7 | 14.9 | 27.3 | 5.7 |
| | MAE | 19.4 | 28.7 | 28.8 | 47.6 | 58.1 | 0.6 | 45.6 | 9.8 | 10.2 | 1.2 |
| | DINO | 66.5 | 77.4 | 62.3 | 80.5 | 69.2 | 1.0 | 89.2 | 20.5 | 43.5 | 9.4 |
| | CLIP | 62.3 | 79.3 | 61.0 | 56.4 | 61.0 | 9.6 | 89.8 | 39.9 | 57.6 | 13.3 |
| AC w/ C | Supervised | **78.5** | **82.6** | 65.5 | 84.2 | 70.8 | 1.7 | 68.4 | 18.2 | 38.5 | 8.7 |
| | MoCo-v3 | 61.0 | 79.9 | 62.1 | 83.9 | **72.7** | 1.4 | 81.1 | 14.9 | 30.6 | 9.7 |
| | MAE | 24.2 | 28.7 | 29.4 | 59.3 | 61.8 | 0.5 | 52.9 | 10.2 | 12.0 | 1.8 |
| | DINO | 67.9 | 74.5 | 61.8 | 83.3 | 69.1 | 0.9 | 90.2 | 21.7 | 46.8 | 14.6 |
| | CLIP | 61.3 | 79.9 | 61.1 | **87.7** | 68.9 | 11.7 | **92.8** | 42.9 | **61.8** | 16.9 |
| AC w/o C | Supervised | 69.9 | 79.3 | 60.5 | 79.6 | 66.8 | 2.7 | 57.5 | 18.5 | 45.1 | 20.9 |
| | MoCo-v3 | 47.2 | 77.1 | 55.0 | 81.9 | 71.7 | 1.3 | 62.1 | 9.8 | 30.1 | 19.6 |
| | MAE | 28.1 | 30.3 | 26.0 | 59.2 | 56.2 | 1.6 | 43.7 | 8.1 | 17.8 | 5.7 |
| | DINO | 53.5 | 72.8 | 49.3 | 81.4 | 69.0 | 0.9 | 77.7 | 11.4 | 36.0 | 21.9 |
| | CLIP | 57.5 | 75.4 | 60.0 | 74.9 | 56.8 | 15.3 | 88.9 | **44.8** | 58.9 | **28.6** |
| Affinity Prop | Supervised | 21.8 | 52.9 | 36.7 | 44.5 | 44.3 | 4.2 | 42.9 | 12.1 | 32.2 | - |
| | MoCo-v3 | 16.9 | 48.9 | 33.2 | 41.3 | 46.0 | 6.1 | 49.5 | 11.9 | 18.2 | - |
| | MAE | 17.7 | 26.2 | 22.8 | 47.0 | 42.5 | 5.4 | 44.7 | 9.8 | 9.2 | - |
| | DINO | 28.1 | 45.0 | 31.9 | 40.6 | 45.5 | 4.0 | 58.8 | 15.9 | 26.5 | - |
| | CLIP | 27.7 | 50.9 | 32.5 | 48.8 | 43.6 | **16.1** | 75.7 | 36.6 | 41.6 | - |
| HDBSCAN | Supervised | 71.9 | 66.4 | 54.7 | 71.3 | 47.6 | 3.0 | 62.3 | 13.8 | 28.2 | 10.2 |
| | MoCo-v3 | 49.4 | 62.0 | 49.9 | 75.6 | 48.2 | 2.9 | 75.4 | 11.2 | 22.3 | 6.8 |
| | MAE | 2.5 | 3.5 | 4.9 | 40.4 | 30.6 | 0.4 | 23.7 | 3.0 | 4.9 | 2.5 |
| | DINO | 55.7 | 57.7 | 50.7 | 74.5 | 44.9 | 2.2 | 83.8 | 15.4 | 32.5 | 9.4 |
| | CLIP | 49.4 | 61.0 | 48.0 | 84.5 | 43.2 | 13.0 | 88.5 | 32.0 | 46.2 | 11.9 |

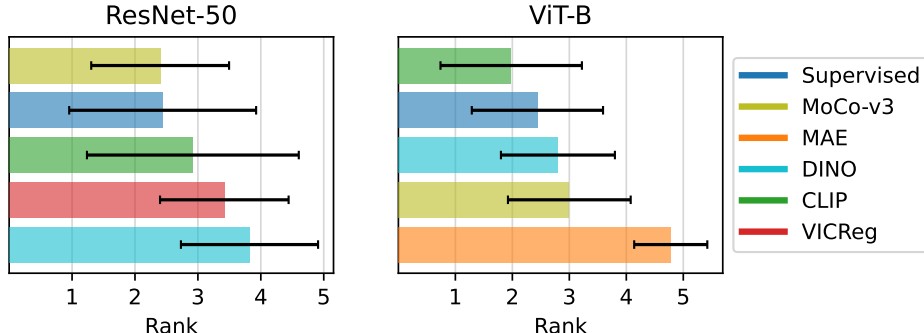

Figure 1: **Average SSL Encoder rank** (lower is better). The average rank of each tested SSL encoder plotted with ± 1 standard deviation. For both the ResNet-50 and ViT-B backbones an SSL encoder in general results in the best clustering. It is worth noting that the supervised method also in general produces good clusters.

ageNet, the gap between the supervised and self-supervised methods is especially noticeable, with a difference of nearly 14 percentage points with the ResNet-50 between the supervised method and the best self-supervised method (CLIP). However, for MNIST and Fashion-MNIST, we find the SSL encoders are much more competitive, with the contrastive MoCo-v3 encoder achieving the highest AMI in all but one case (MNIST with ViT-B, where CLIP is the best encoder), and the supervised network outperformed by multiple SSL encoders. For the smaller fine-grained datasets (FGVC Aircraft, Oxford Flowers, and NABirds) as well as SVHN we find that the multi-modal CLIP encoder achieves the best performance for both ResNet-50 and ViT-B. The supervised network performs particularly poorly on Oxford Flowers (around 10 percentage points worse than the SSL networks). It is worth noting also that the performance on the SVHN dataset is dramatically lower than all other datasets. We believe this is due to the very large intra-class diversity for each digit and small inter-class diversity among digits, originating from the different colored house walls and several digits being visible in each image. In comparison, the images in Oxford Flowers have perceptually less variability within classes, and the clustering has much higher agreement with the annotations for this dataset. Lastly, we find that most combinations of encoders and clustering methods perform poorly on iNaturalist-21, due to the large number of considered species (10,000) spanning the entire tree of life (Van Horn et al., 2021). The exception is AC with unknown amount of clusters where performance is dramatically higher, reaching an AMI of 28.6%.

### 4.1 COMPARISON OF SSL ENCODERS

In order to directly compare the different SSL-trained encoders, we rank each encoder across datasets and clustering methods, shown in Fig. 1. We find that there is a clear ranking for both backbone architectures, with MoCo-v3 performing the best for ResNet-50, and CLIP best for ViT-B. It is worth noting that with a ResNet-50 backbone the CLIP method performs poorly, even though it is trained on a much larger dataset than ImageNet-1k. We also find that the supervised baseline is the second-best for both backbones. It is also noticeable that the DINO self-distillation approach performs well using a ViT-B backbone, but very poorly with ResNet-50 (the same trend as seen for CLIP); this corroborates the findings of Vaze et al. (2022). Lastly, the MAE encoder performed particularly poorly across all datasets we considered. This finding is congruent with the observation that MAE-trained models possess details about the pixel-level contents of the stimulus, but need fine-tuning to perform well at whole-image classification (He et al., 2022).

### 4.2 COMPARISON OF CLUSTERING METHODS

We compared the performance of the clustering methods by ranking each method for each combination of SSL encoder and dataset, shown in Fig. 2. From the average ranking, it is immediately obvious that the best performing clustering method across both backbones is Agglomerative Clustering with the number of clusters known *a priori*. However, we find that Agglomerative Clustering

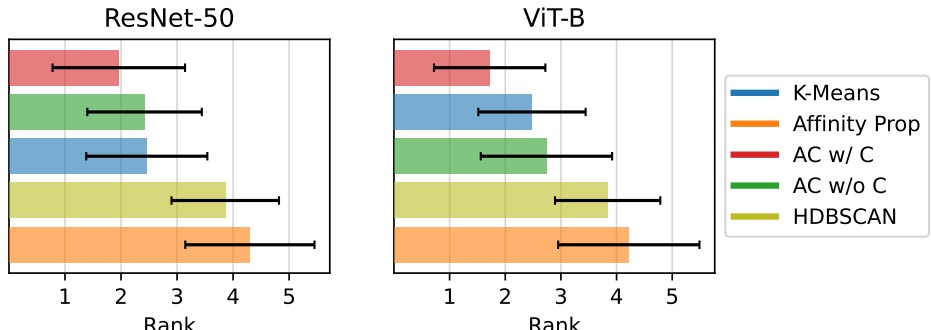

Figure 2: **Average clustering method rank** (lower is better). The average rank of each tested clustering method plotted with ± 1 standard deviation. The Agglomerative Clustering method performs very well, whether the number of cluster are known *a priori* (Red) or not (Green).

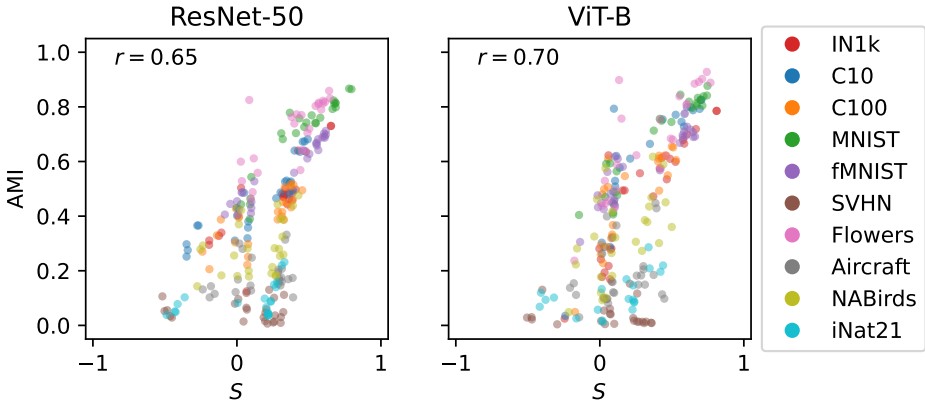

Figure 3: **AMI–Silhouette scatter plots.** The AMI and silhouette score, $S$, of each SSL encoder and clustering method combination are plotted against each other across all datasets, per backbone. We find that there is strong correlation between the two metrics.

with an unknown number of clusters is very competitive, outperforming K-Means when using a ResNet-50 backbone. We also find that HDBSCAN and Affinity Propagation are consistently the worst performing clustering methods we considered.

### 4.3 EFFECT OF DATASET GRANULARITY

We observe that the performance varies on the smaller fine-grained datasets. While seemingly arbitrary, we find that the performance correlates with how fine-grained the datasets are when considering the proposed granularity measure from Cui et al. (2019). Specifically we find that FGVC Aircraft is the most challenging dataset, matching the finding by Cui et al. (2019) that it is the most fine-grained dataset of the ones considered, while NABirds and Oxford Flowers gradually becomes more coarse-grained, and easier to correctly cluster. Similarly, we find that the large scale iNaturalist-21 dataset is in general a very hard dataset. These observations echo the recent results from Cole et al. (2022), where it was determined that current SSL methods are not suitable for fine-grained tasks.

### 4.4 CORRELATION BETWEEN AMI AND SILHOUETTE SCORE

In the prior analysis we have focused on the AMI metric, which provides a performance measure by directly comparing the predicted clusters with the ground truth clusters. However, in the context of SSL this is problematic since there is no ground truth available. Therefore, the intrinsic Silhouette

Table 3: **AMI and Silhouette score correlations.** We compute the Pearson correlation between the AMI and $S$ metrics for each dataset and each clustering method.

(a) **Per-dataset correlation coefficients.** A clear correlation is determined for the ImageNet, CIFAR, and MNIST style datasets. In contrast, the majority of the fine-grained datasets have a weaker correlation, except for Oxford Flowers, while SVHN is completely uncorrelated.

| Backbone | IN1k | C10 | C100 | MNIST | fMNIST | SVHN | Flowers | Aircraft | NABirds | iNat21 |
|---|---|---|---|---|---|---|---|---|---|---|
| ResNet-50 | 0.82 | 0.89 | 0.76 | 0.95 | 0.97 | $-0.10$ | 0.75 | 0.28 | 0.40 | 0.40 |
| ViT-B | 0.91 | 0.87 | 0.82 | 0.95 | 0.91 | $-0.09$ | 0.84 | 0.33 | 0.57 | 0.51 |

(b) **Per-clusterer correlation coefficients.** A strong correlation is determined for Agglomerative Clustering and HDBSCAN. Affinity Propagation exhibit the weakest correlation with a ResNet-50 backbone, but achieve a much stronger correlation with a ViT-B backbone.

| Backbone | K-Means | Affinity Prop | AC w/ C | AC w/o C | HDBSCAN |
|---|---|---|---|---|---|
| ResNet-50 | 0.61 | 0.15 | 0.91 | 0.86 | 0.94 |
| ViT-B | 0.60 | 0.55 | 0.82 | 0.66 | 0.94 |

metric, $S$, calculated from just the predicted clusters is potentially valuable for evaluation of SSL encoders. However, it is unclear whether AMI and $S$ are correlated. Therefore, we compare the AMI and $S$ of each SSL encoder and clustering method across all datasets for both backbones, see Fig. 3, and compute the Pearson correlation coefficient ($r$) across all datapoints. Here we find that AMI and $S$ are in general strongly correlated, with low AMI scores having correspondingly low Silhouette scores. When looking at per-dataset $r$ values, see Table 3a, we find that strongest correlations are obtained for ImageNet, CIFAR-10, CIFAR-100, MNIST and Fashion MNIST. However, for all fine-grained datasets (except Oxford Flowers) the strength of the correlation drops dramatically. For SVHN the metrics are not correlated at all, since AMI was very low across all models, irrespective of $S$. Looking at the per-clustering method results, see Table 3b, we find that the AMI and $S$ metrics are strongly correlated for the Agglomerative Clustering and HDBSCAN methods, while Affinity Propagation is very weakly correlated when using a ResNet-50 backbone.

Therefore, we can conclude that the Silhouette score can be a good proxy when ground truth labels are not available, but that the effectiveness of the proxy diminishes as the data becomes more fine-grained and further from the training domain.

## 5 CONCLUSION

We have empirically investigated how well the feature embeddings produced by pretrained networks can be clustered in a zero-shot setting, exploring two different architectures trained using one of six different methodologies (one supervised, five self-supervised), on 10 different datasets, using five classic clustering methods.

We find that it is possible to create well-defined clusters across nearly all tested datasets, even for notoriously hard fine-grained datasets such as NABirds. In many cases, the performance on novel datasets was equal or comparable to that on the in-domain test set of ImageNet-1k. Agglomerative Clustering is found to be the consistently strongest clusterer when the number of clusters is known *a priori*, and also the strongest choice when the number of classes are not known, provided its distance threshold is tuned on a labelled dataset. In contrast, there is not a single overall best SSL paradigm. Instead, we find the contrastive MoCo-V3 method is the best method with a ResNet-50, whereas the multi-modal CLIP approach is the strongest when using a ViT-B backbone.

To cluster embeddings of a novel dataset, we recommend reducing the dimensionality down using UMAP (we used 50d, any amount of dimensions in the range 5–100 should work), then applying Agglomerative Clustering on the reduced embeddings. We also show promising results that the Silhouette score can be used to evaluate SSL methods for clustering when no ground truth is available.

We believe these results shed an important light on the capabilities of SSL trained feature encoders, and highlight that they in many cases can produce meaningful clusters on new datasets without any additional tuning.

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

Table 4: **Predicted number of clusters.** For each clusterer, we report the number of clusters generated. We report the ground-truth number of classes in the dataset (Num targets), information which the clusterer was blinded to.

(a) **ResNet-50 backbone.**

|  | Encoder | IN1k | C10 | C100 | MNIST | fMNIST | SVHN | Flowers | Aircraft | NABirds | iNat21 |
|---|---|---|---|---|---|---|---|---|---|---|---|
|  | Num targets | 1000 | 10 | 100 | 10 | 10 | 10 | 102 | 100 | 555 | 10000 |
| AC w/o C | Supervised | 147 | 12 | 22 | 16 | 15 | 33 | 21 | 9 | 38 | 76 |
| AC w/o C | MoCo-v3 | 63 | 14 | 19 | 14 | 15 | 32 | 25 | 8 | 28 | 72 |
| AC w/o C | VICReg | 70 | 16 | 22 | 16 | 15 | 30 | 29 | 7 | 29 | 69 |
| AC w/o C | DINO | 84 | 27 | 38 | 26 | 20 | 54 | 54 | 21 | 41 | 64 |
| AC w/o C | CLIP | 70 | 28 | 41 | 32 | 27 | 61 | 57 | 31 | 47 | 44 |
| Affinity Prop | Supervised | - | 392 | 459 | 401 | 338 | 1041 | 340 | 385 | 974 | - |
| Affinity Prop | MoCo-v3 | - | 295 | 348 | 296 | 194 | 839 | 254 | 132 | 536 | - |
| Affinity Prop | VICReg | 1394 | 320 | 346 | 318 | 231 | 807 | 234 | 130 | 529 | - |
| Affinity Prop | DINO | - | 741 | 872 | 754 | 623 | 3051 | 636 | 371 | 1507 | - |
| Affinity Prop | CLIP | - | 286 | 366 | 288 | 207 | 696 | 224 | 132 | 632 | - |
| HDBSCAN | Supervised | 1181 | 228 | 201 | 81 | 178 | 617 | 181 | 98 | 526 | 1617 |
| HDBSCAN | MoCo-v3 | 1302 | 222 | 232 | 81 | 214 | 544 | 140 | 114 | 414 | 1685 |
| HDBSCAN | VICReg | 1212 | 225 | 242 | 83 | 180 | 594 | 156 | 115 | 563 | 1687 |
| HDBSCAN | DINO | 1163 | 265 | 250 | 87 | 188 | 678 | 153 | 109 | 376 | 1683 |
| HDBSCAN | CLIP | 1072 | 276 | 252 | 57 | 225 | 631 | 160 | 83 | 462 | 1328 |

(b) **ViT-B backbone.**

|  | Encoder | IN1k | C10 | C100 | MNIST | fMNIST | SVHN | Flowers | Aircraft | NABirds | iNat21 |
|---|---|---|---|---|---|---|---|---|---|---|---|
|  | Num targets | 1000 | 10 | 100 | 10 | 10 | 10 | 102 | 100 | 555 | 10000 |
| AC w/o C | Supervised | 226 | 14 | 25 | 13 | 16 | 31 | 20 | 8 | 36 | 78 |
| AC w/o C | MoCo-v3 | 93 | 16 | 20 | 14 | 11 | 13 | 23 | 6 | 18 | 28 |
| AC w/o C | MAE | 131 | 26 | 30 | 35 | 27 | 68 | 22 | 10 | 60 | 218 |
| AC w/o C | DINO | 90 | 8 | 14 | 9 | 9 | 6 | 33 | 5 | 18 | 17 |
| AC w/o C | CLIP | 111 | 30 | 71 | 35 | 55 | 125 | 58 | 40 | 78 | 79 |
| Affinity Prop | Supervised | 1456 | 164 | 274 | 261 | 198 | 712 | 223 | 168 | 426 | - |
| Affinity Prop | MoCo-v3 | 1274 | 205 | 293 | 292 | 185 | 756 | 234 | 101 | 440 | - |
| Affinity Prop | MAE | 2009 | 422 | 461 | 400 | 304 | 840 | 355 | 170 | 948 | - |
| Affinity Prop | DINO | 1090 | 246 | 330 | 284 | 201 | 698 | 222 | 98 | 420 | - |
| Affinity Prop | CLIP | 738 | 196 | 314 | 250 | 192 | 617 | 174 | 108 | 566 | - |
| HDBSCAN | Supervised | 1123 | 118 | 209 | 100 | 209 | 594 | 174 | 92 | 502 | 1325 |
| HDBSCAN | MoCo-v3 | 1145 | 105 | 235 | 85 | 171 | 548 | 162 | 97 | 456 | 1592 |
| HDBSCAN | MAE | 133 | 19 | 21 | 17 | 14 | 51 | 52 | 10 | 50 | 40 |
| HDBSCAN | DINO | 1142 | 144 | 226 | 77 | 215 | 634 | 152 | 110 | 477 | 1555 |
| HDBSCAN | CLIP | 968 | 138 | 237 | 40 | 231 | 630 | 128 | 142 | 449 | 1390 |

Jinghao Zhou, Chen Wei, Huiyu Wang, Wei Shen, Cihang Xie, Alan Yuille, and Tao Kong. ibot: Image bert pre-training with online tokenizer, 2022.

Xingzhi Zhou and Nevin L. Zhang. Deep clustering with features from self-supervised pretraining, 2022.

# A APPENDIX

## A.1 PREDICTED NUMBER OF CLUSTERS

We report the predicted number of clusters for the three clusterers which do not require a number of clusters to be provided to the clusterer.

As shown in Table 4, the number of clusters predicted is typically a consistent order of magnitude for a given clusterer and dataset, irrespective of the encoder used. However there is great variability between clusterers. Affinity propagation predicted a couple of hundred clusters, irrespective of the dataset. Agglomerative clustering predicted the fewest clusters, even predicting only in the order of

Table 5: **Silhouette scores, with ResNet-50 backbone.** We report the Silhouette score, on each tested dataset (see Table 1) for each combination of SSL encoder and clustering method. The performance of Agglomerative Clustering is shown twice, either using the ground-truth number of classes as the number of clusters to generate (AC w/ C), or predicting the number of clusters (AC w/o C). The hyperparameters of the clustering methods are only tuned on ImageNet-1k (IN1k). The best combination of SSL encoder and clustering method per dataset and backbone is highlighted in **bold**, while the best SSL encoder per clustering method is underlined. We also present the Silhouette scores attained for the embeddings using the "ground-truth" classes as per the dataset annotation (G.T.).

| | Encoder | IN1k | C10 | C100 | MNIST | fMNIST | SVHN | Flowers | Aircraft | NABirds | iNat21 |
|---|---|---|---|---|---|---|---|---|---|---|---|
| **G.T.** | Supervised | 0.57 | 0.46 | 0.38 | 0.58 | 0.57 | 0.23 | 0.45 | 0.29 | 0.31 | 0.24 |
| | MoCo-v3 | 0.34 | 0.42 | 0.35 | 0.72 | 0.58 | 0.22 | 0.52 | 0.30 | 0.28 | 0.22 |
| | VICReg | 0.34 | 0.34 | 0.33 | 0.62 | 0.58 | 0.20 | 0.55 | 0.30 | 0.25 | 0.22 |
| | DINO | 0.33 | 0.32 | 0.30 | 0.48 | 0.53 | 0.20 | 0.58 | 0.29 | 0.26 | 0.21 |
| | CLIP | 0.34 | 0.34 | 0.30 | 0.55 | 0.44 | 0.20 | 0.63 | 0.33 | 0.29 | 0.19 |
| **K-Means** | Supervised | **0.65** | **0.49** | 0.39 | 0.69 | 0.61 | 0.32 | 0.47 | 0.30 | 0.32 | 0.24 |
| | MoCo-v3 | 0.38 | 0.47 | 0.37 | **0.80** | **0.62** | **0.33** | 0.55 | 0.31 | 0.29 | 0.22 |
| | VICReg | 0.38 | 0.38 | 0.34 | 0.69 | **0.62** | 0.31 | 0.60 | 0.31 | 0.27 | 0.22 |
| | DINO | 0.35 | 0.39 | 0.31 | 0.54 | 0.57 | 0.30 | 0.57 | 0.31 | 0.27 | 0.22 |
| | CLIP | 0.03 | 0.05 | 0.01 | 0.10 | 0.12 | 0.05 | 0.09 | 0.02 | 0.03 | 0.00 |
| **AC w/ C** | Supervised | **0.65** | 0.47 | 0.37 | 0.68 | 0.59 | 0.27 | 0.45 | 0.29 | 0.30 | 0.23 |
| | MoCo-v3 | 0.36 | 0.44 | 0.33 | 0.78 | 0.60 | 0.26 | 0.57 | 0.29 | 0.26 | 0.21 |
| | VICReg | 0.36 | 0.35 | 0.32 | 0.66 | 0.61 | 0.26 | 0.61 | 0.30 | 0.24 | 0.22 |
| | DINO | 0.33 | 0.36 | 0.28 | 0.55 | 0.55 | 0.21 | 0.60 | 0.28 | 0.25 | 0.20 |
| | CLIP | 0.34 | 0.37 | 0.28 | 0.68 | 0.49 | 0.25 | 0.64 | 0.34 | 0.28 | 0.19 |
| **AC w/o C** | Supervised | 0.48 | 0.46 | **0.45** | 0.48 | 0.55 | 0.20 | 0.44 | **0.39** | **0.43** | 0.31 |
| | MoCo-v3 | 0.32 | 0.40 | 0.39 | 0.67 | 0.56 | 0.22 | 0.49 | 0.36 | 0.32 | 0.26 |
| | VICReg | 0.32 | 0.33 | 0.38 | 0.58 | 0.55 | 0.20 | 0.50 | 0.39 | 0.32 | 0.27 |
| | DINO | 0.32 | 0.27 | 0.31 | 0.41 | 0.50 | 0.19 | 0.59 | 0.29 | 0.30 | 0.28 |
| | CLIP | 0.41 | 0.30 | 0.33 | 0.43 | 0.40 | 0.20 | **0.65** | 0.34 | 0.39 | **0.33** |
| **Affinity Prop** | Supervised | - | 0.00 | 0.01 | 0.03 | 0.01 | 0.01 | 0.03 | −0.01 | 0.01 | - |
| | MoCo-v3 | - | 0.08 | 0.08 | 0.09 | 0.10 | 0.07 | 0.10 | 0.09 | 0.08 | - |
| | VICReg | 0.07 | 0.07 | 0.08 | 0.09 | 0.10 | 0.07 | 0.11 | 0.10 | 0.08 | - |
| | DINO | - | −0.01 | −0.01 | 0.00 | 0.01 | −0.01 | 0.03 | 0.01 | 0.00 | - |
| | CLIP | - | 0.08 | 0.07 | 0.09 | 0.10 | 0.07 | 0.12 | 0.10 | 0.07 | - |
| **HDBSCAN** | Supervised | 0.42 | −0.27 | −0.01 | 0.31 | 0.03 | −0.48 | 0.14 | −0.19 | −0.25 | −0.41 |
| | MoCo-v3 | −0.11 | −0.28 | −0.11 | 0.52 | 0.00 | −0.52 | 0.38 | −0.18 | −0.02 | −0.43 |
| | VICReg | −0.12 | −0.35 | −0.14 | 0.47 | 0.04 | −0.50 | 0.45 | −0.15 | −0.28 | −0.43 |
| | DINO | −0.19 | −0.34 | −0.24 | 0.32 | −0.05 | −0.46 | 0.41 | −0.24 | −0.11 | −0.49 |
| | CLIP | −0.19 | −0.35 | −0.19 | 0.37 | −0.08 | −0.45 | 0.40 | 0.04 | −0.24 | −0.36 |

100 clusters for Imagenet-1k, the dataset the encoders were trained on. HDBSCAN varied more in the number of clusters it predicted, with around the right number of classes being predicted for the datasets which were comprised of at least 100 classes.

## A.2 SILHOUETTE SCORES

We report the Silhouette scores for each clustering of the test datasets, shown for ResNet-50 architectures in Table 5 and ViT-B architectures in Table 6.

Our results on the Silhouette score are broadly in line with our main finding on the AMI between clusterings and annotation targets, reported in §4. For both the ResNet-50 and ViT-B encoders, the supervised model has the highest Silhouette score by a large margin of 0.25–0.3, but otherwise the clustering quality across the encoders is very similar, achieving similar Silhouette scores to each other. There are some exceptions to this, such as the Silhouette scores for MAE which are near 0, illustrating the intrinsically-poor quality of the clusters it exhibited and hence it is not well-suited to this task.

Despite the very low AMI scores, we observe the Silhouette scores for SVHN are generally comparable to the Silhouette scores of the other datasets. We believe this is due to the heterogeneity within

Table 6: **Silhouette scores, with ViT-B backbone.** As for Table 5, except for encoders with ViT-B backbones instead of ResNet-50.

| | Encoder | IN1k | C10 | C100 | MNIST | fMNIST | SVHN | Flowers | Aircraft | NABirds | iNat21 |
|---|---|---|---|---|---|---|---|---|---|---|---|
| G.T. | Supervised | 0.74 | 0.70 | 0.50 | 0.68 | 0.57 | 0.23 | 0.46 | 0.27 | 0.32 | 0.24 |
| | MoCo-v3 | 0.43 | 0.57 | 0.44 | 0.66 | 0.60 | 0.27 | 0.53 | 0.29 | 0.27 | 0.21 |
| | MAE | −0.22 | 0.02 | 0.02 | 0.04 | 0.03 | 0.03 | 0.04 | 0.03 | 0.02 | −0.07 |
| | DINO | 0.52 | 0.58 | 0.43 | 0.67 | 0.62 | 0.36 | 0.64 | 0.30 | 0.35 | 0.23 |
| | CLIP | 0.42 | 0.53 | 0.41 | 0.65 | 0.50 | 0.21 | 0.73 | 0.39 | 0.38 | 0.19 |
| K-Means | Supervised | **0.81** | **0.71** | 0.51 | 0.71 | 0.58 | 0.30 | 0.51 | 0.29 | 0.33 | 0.24 |
| | MoCo-v3 | 0.53 | 0.65 | 0.46 | 0.70 | 0.59 | 0.32 | 0.60 | 0.31 | 0.29 | 0.23 |
| | MAE | 0.03 | 0.07 | 0.05 | 0.12 | 0.16 | 0.10 | 0.06 | 0.05 | 0.04 | 0.02 |
| | DINO | 0.59 | 0.59 | 0.45 | 0.67 | 0.59 | **0.36** | 0.69 | 0.30 | 0.34 | 0.23 |
| | CLIP | 0.06 | 0.10 | 0.06 | 0.12 | 0.12 | 0.05 | 0.13 | 0.05 | 0.04 | 0.01 |
| AC w/C | Supervised | **0.81** | 0.70 | 0.50 | **0.74** | 0.65 | 0.24 | 0.49 | 0.26 | 0.31 | 0.24 |
| | MoCo-v3 | 0.49 | 0.60 | 0.41 | 0.72 | 0.61 | 0.26 | 0.60 | 0.27 | 0.25 | 0.20 |
| | MAE | 0.01 | 0.03 | 0.01 | 0.07 | 0.10 | 0.04 | 0.03 | 0.03 | 0.01 | 0.02 |
| | DINO | 0.58 | 0.57 | 0.44 | 0.68 | **0.66** | 0.31 | 0.71 | 0.30 | 0.36 | 0.23 |
| | CLIP | 0.43 | 0.63 | 0.41 | **0.75** | 0.58 | 0.26 | 0.74 | 0.39 | 0.37 | 0.19 |
| AC w/o C | Supervised | 0.60 | 0.69 | **0.53** | 0.64 | 0.56 | 0.23 | 0.42 | 0.37 | 0.46 | 0.35 |
| | MoCo-v3 | 0.37 | 0.53 | 0.47 | 0.62 | 0.63 | 0.27 | 0.46 | **0.45** | 0.41 | 0.42 |
| | MAE | −0.00 | 0.01 | 0.01 | 0.04 | 0.05 | 0.01 | 0.01 | 0.06 | 0.01 | −0.01 |
| | DINO | 0.47 | 0.58 | 0.41 | 0.67 | 0.65 | **0.36** | 0.59 | 0.43 | **0.50** | **0.44** |
| | CLIP | 0.42 | 0.44 | 0.40 | 0.43 | 0.42 | 0.21 | **0.77** | 0.39 | 0.44 | 0.34 |
| Affinity Prop | Supervised | 0.06 | 0.11 | 0.09 | 0.10 | 0.10 | 0.07 | 0.10 | 0.09 | 0.11 | - |
| | MoCo-v3 | 0.07 | 0.09 | 0.09 | 0.09 | 0.10 | 0.07 | 0.11 | 0.10 | 0.09 | - |
| | MAE | 0.02 | 0.02 | 0.02 | 0.04 | 0.03 | 0.03 | 0.04 | 0.03 | 0.02 | - |
| | DINO | 0.08 | 0.09 | 0.08 | 0.09 | 0.10 | 0.07 | 0.12 | 0.09 | 0.08 | - |
| | CLIP | 0.07 | 0.11 | 0.09 | 0.11 | 0.11 | 0.08 | 0.15 | 0.12 | 0.09 | - |
| HDBSCAN | Supervised | 0.67 | 0.35 | 0.17 | 0.37 | −0.02 | −0.47 | 0.25 | −0.21 | −0.16 | −0.32 |
| | MoCo-v3 | 0.17 | 0.12 | 0.03 | 0.42 | 0.07 | −0.51 | 0.42 | −0.17 | −0.17 | −0.42 |
| | MAE | −0.24 | −0.24 | −0.17 | −0.14 | −0.14 | −0.30 | −0.18 | −0.14 | −0.22 | −0.16 |
| | DINO | 0.28 | 0.14 | 0.06 | 0.50 | 0.01 | −0.47 | 0.55 | −0.18 | −0.04 | −0.38 |
| | CLIP | 0.16 | 0.21 | 0.02 | 0.56 | −0.01 | −0.40 | 0.66 | 0.09 | −0.02 | −0.41 |

the classes in SVHN, where house-numbers can be written in different formats, colours, etc., and thus the encoded images can be appropriately grouped together, even if the semantic meaning of the clusters does not correspond to the identity of the digit in the center of the image.

Between the clusterers, K-Means and AC typically achieve the highest Silhouette scores. For HDB-SCAN, the Silhouette scores were often significantly negative. This is because HDBSCAN builds clusters based on transitions in density, and the non-convex clusters that result from this can score poor Silhouette scores (a known caveat to this evaluation metric). For Affinity Propagation, we observe Silhouette scores near 0, indicating the clusters it discovered have high overlap with each other and are of low quality, corresponding to its poor AMI performance.

