# OpenReview forum: "Zero-shot Clustering of Embeddings with Pretrained and Self-Supervised Learning Encoders"
_ICLR.cc/2024/Conference — Submitted to ICLR 2024_

### Official Review · Reviewer_JmUn · 2023-10-30

**Soundness:** 3 good
**Presentation:** 3 good
**Contribution:** 3 good
**Rating:** 5
**Confidence:** 4

**Summary:**

The authors analyze the performance of unsupervised clustering on the feature space of five self-supervised learning methods (one for each major SSL paradigm) and a supervised learning baseline. Two architectures for feature extraction and five clustering techniques. Each possible combination of these three aspects (training method, clustering technique and backbone network) is tested on 10 datasets with varying levels of classification granularity; clustering performance is evaluated through adjusted mutual information and the silhouette score.

**Strengths:**

- The writing is easy to follow
- This analysis has not been done before and will be of good use for the community of practitioners looking into clustering methods for pre-trained feature spaces
- Names the concept "Zero-shot Clustering", which while not being a new idea (most clustering methods are by themselves zero-shot) it is helpful to differentiate from representation learners within-domain.
- The test on correlation between silhouette and adjusted mutual information makes a strong case for starting data analysis from clusterization in unsupervised applications.

**Weaknesses:**

- This is the first instance of the "zero shot clustering" task, yet the task has not been formally defined in the manuscript. A formal definition should for example define that the class sets between the feature extractor's training set and the set being clustered as disjoint; this is important to avoid problems like the unfairness of comparing CLIP embeddings on equal grounds.
- On a related/followup note: Including comparisons against CLIP on the same grounds is unfair to other feature extractors and detracts from the "zero-shot" nature of the task. Because it is not possible to determine if the classes (or even the samples themselves) were present the the CLIP training set, the results on CLIP should be included as extra information and treated as such. The current discussion is hindered by comparing against CLIP directly as readers are not able to properly judge the quality of the feature extractor's generalization.
- The hyper parameter sweep (second stage) considers parameters only for Agglomerative Clustering (AC), with only HDBSCAN also getting any parameter evaluation (albeit for a single parameter choice). K-means could have it's distance metric changed as well as have an alternative formulation with automatic K (elbow method or even using silhouette score itself). Affinity Propagation also has hyper parameters that could be candidate for a sweep. The lack of experimentation with the other clustering methods raises concerns regarding the final results as AC ends up being the recommended method.

Other notes:
- First paragraph is missing a period.
- Default parameters may change over time on packages such as scikit-learn. It is better to include all of the necessary information for reproduction (either on th main manuscript or on a supplementary material file).
- Figure 1 and 2 are hard to read due to change in order. It might also help to include the actual average ranking number on the plot.
- Sec. 4.1: authors did not mention which findings from Vaze et al. are corroborated (include the insight instead of only the result)
- Conclusion: Performance was not equal or comparable on fine-grained datasets

**Questions:**

1. Excluding CLIP, what would the authors recommend between using CNN or Transformer-based architectures? It is important to consider that a ViT-B is not computationally comparable to a ResNet50 (which is lighter by 3~4x in FLOPs and parameters).

**Details Of Ethics Concerns:**

Only publicly available models and datasets are used.

---

> ### Author Response · Authors · 2023-11-23
>
> We appreciate the time and effort the reviewer has put into analyzing our work, that they found it easy to follow, and that they found it interesting.
>
> > This is the first instance of the "zero shot clustering" task, yet the task has not been formally defined in the manuscript. A formal definition should for example define that the class sets between the feature extractor's training set and the set being clustered as disjoint; this is important to avoid problems like the unfairness of comparing CLIP embeddings on equal grounds.
>
> We agree that the omission of a clear definition of “zero-shot clustering” is unfortunate and we will make sure that this will be included in a future version.
>
> > On a related/followup note: Including comparisons against CLIP on the same grounds is unfair to other feature extractors and detracts from the "zero-shot" nature of the task. Because it is not possible to determine if the classes (or even the samples themselves) were present the the CLIP training set, the results on CLIP should be included as extra information and treated as such. The current discussion is hindered by comparing against CLIP directly as readers are not able to properly judge the quality of the feature extractor's generalization.
>
> This is a fair point. We are currently aiming to include results using the OpenCLIP model, for which we know what data the model was trained on, and therefore can better interpret the results and how they fit within the “zero-shot” scenario.
>
> > The hyper parameter sweep (second stage) considers parameters only for Agglomerative Clustering (AC), with only HDBSCAN also getting any parameter evaluation (albeit for a single parameter choice). K-means could have it's distance metric changed as well as have an alternative formulation with automatic K (elbow method or even using silhouette score itself). Affinity Propagation also has hyper parameters that could be candidate for a sweep. The lack of experimentation with the other clustering methods raises concerns regarding the final results as AC ends up being the recommended method.
>
> Our preliminary analysis showed that tuning the HDBSCAN metric distance did not have a notable impact on performance, hence we prioritized tuning AC over HDBSCAN. After the submission, for completeness-sake we extended our full hyperparameter sweep over the hyperparameters of Affinity Propagation and HDBSCAN, and confirmed that their final performance was not notably affected by this further optimization.
>
> > Other notes: [...]
>
> We will make sure to address all the mentioned points. Regarding the reference to Vaze et al. in Sec 4.1 we will make sure to include the insight, which is: datapoint clusters are more well defined with a DINO ViT backbone than DINO ResNet backbone
>
> > Excluding CLIP, what would the authors recommend between using CNN or Transformer-based architectures? It is important to consider that a ViT-B is not computationally comparable to a ResNet50 (which is lighter by 3~4x in FLOPs and parameters).
>
> Based on our findings we would recommend using contrastive-learning based approaches, given that both MoCo and CLIP are contrastive-based methods. We do also find that there isn’t a single specific method that is good for both backbone types. Surprisingly, we also find that the supervised encoders are in general very good for clustering, which was not originally expected. These results indicate that there is no single best method at the moment, and that our understanding of SSL methods is not fully complete.
>
> With regards to CNN vs Transformer, our experimental paradigm was not set up to investigate this. As the reviewer points out, a ViT-B requires notably more compute than ResNet50. Our choice of architectures was driven by which pretrained models were readily available in accompaniment to prominent SSL papers. Our results indicate that ViT-B performs better than ResNet-50 when the new dataset is in-domain or a similar domain to the training data, but there is little difference between them for out-of-domain datasets. However as we only investigated two backbones, we do not have sufficient evidence to make any strong conclusions on this point.

---

### Official Review · Reviewer_B14Z · 2023-10-30

**Soundness:** 2 fair
**Presentation:** 2 fair
**Contribution:** 1 poor
**Rating:** 1
**Confidence:** 5

**Summary:**

This work explores the image clustering ability of pretrained model. The study focus on 6 supervised/self-supervised methods with ViT-B and ResNet-50.

**Strengths:**

- The experiments are somewhat extensive, covering a range of different models and datasets.

**Weaknesses:**

- Is the testing set truly out-of-distribution? For a lot of the dataset tested, it heavily overlap with ImageNet images or CLIP training images. I think it is ok to say "the work benchmarked a lot of datasets" but it is not accurate to say "the work benchmarked a lot of out-of-distribution datasets"

- The metric used. There has been a long-standing image clustering and deep image clustering community (e.g.https://github.com/zhoushengisnoob/DeepClustering) that measures the accuracy of clustering directly. This has became the most important metric to measure clustering performance.

- Is this work truly the **first** investigation into zero-shot clustering of SSL feature?: Since 2020, there has been lots of work focusing on image-clustering based on a pretrained (SSL) image encoder. The papers have shown that (1) zero-shot clustering can have non-trivial performance (similar to the result of this paper) (2) Performance can be further improved with different finetuning methods. I think it is important to acknowledge the previous works and reconsider the novelty of this work.

- Some issues with definition and terminology: CLIP is not self-supervised, it is rather supervised/language-supervised/weakly-supervised.
MoCo/DINO/VICReg have a lot in common. Last year's ICLR best paper candidate has shown a strong duality between the methods, so it would be more accurate to just say they are contrastive.

- Writing and Presentation: I think the work can be improved a lot, with the current format.

**Questions:**

Please see weakness

---

> ### Author Response · Authors · 2023-11-23
>
> We appreciate the time the reviewer has put into reviewing our work.
>
> > Is the testing set truly out-of-distribution? For a lot of the dataset tested, it heavily overlap with ImageNet images or CLIP training images. I think it is ok to say "the work benchmarked a lot of datasets" but it is not accurate to say "the work benchmarked a lot of out-of-distribution datasets"
>
> We agree with the reviewer that not all of the datasets are necessarily out-of-distribution, as there is partial overlap between the domains Therefore we will reformulate our analysis and instead consider three kinds of dataset groups:
> - In distribution: ImageNet-1k, ImageNet-V2 (new), CIFAR-10, CIFAR-100
> - Fine-grained: Flowers102, FGVC aircraft, NABirds, iNaturalist
> - OOD: ImageNet-R (new), MNIST, fMNIST, SVHN
>
> > The metric used. There has been a long-standing image clustering and deep image clustering community (e.g.https://github.com/zhoushengisnoob/DeepClustering) that measures the accuracy of clustering directly. This has became the most important metric to measure clustering performance.
>
> We politely disagree that “accuracy” is the most important metric for clustering performance. While it may have become more commonly used, classic metrics such as Adjusted Mutual Information are still widely used and of great value. We also want to highlight that in the recent survey [[1](https://arxiv.org/abs/2206.07579)] (a paper highlighted on the GitHub page linked to us by the reviewer), Normalized Mutual Information (NMI), Adjusted Rand Index (ARI), and Accuracy are mentioned as the most common metrics. NMI is an arguably worse version of AMI as it does not correct for random chance agreement levels which vary with the number of clusters used, and therefore it makes sense to instead use AMI.ARI is highly redundant with AMI, hence we did not include it in the paper despite computing it in the analysis. Moreover, accuracy requires a 1-to-1 mapping via the Hungarian algorithm, which is not possible when the clusterer is allowed to determine the number of clusters in its output.
>
> > Is this work truly the first investigation into zero-shot clustering of SSL feature?: Since 2020, there has been lots of work focusing on image-clustering based on a pretrained (SSL) image encoder. The papers have shown that (1) zero-shot clustering can have non-trivial performance (similar to the result of this paper) (2) Performance can be further improved with different finetuning methods. I think it is important to acknowledge the previous works and reconsider the novelty of this work.
>
> While there have been prior methods that combine SSL-features with classical clustering, such as [[2](https://openaccess.thecvf.com/content/CVPR2022/html/Ronen_DeepDPM_Deep_Clustering_With_an_Unknown_Number_of_Clusters_CVPR_2022_paper.html)], these methods have optimised the latent space for clustering first, resulting in the procedure not being zero-shot. We would greatly appreciate any references to the reviewer’s two claims and will accordingly change our manuscript.
>
> > Some issues with definition and terminology: CLIP is not self-supervised, it is rather supervised/language-supervised/weakly-supervised. MoCo/DINO/VICReg have a lot in common. Last year's ICLR best paper candidate has shown a strong duality between the methods, so it would be more accurate to just say they are contrastive
>
> We agree that CLIP can be categorised as weakly supervised learning. However, as the model is learnt with a contrastive learning loss-function, we also argue that it is fair to categorise it as “cross-domain contrastive learning” and thereby as self-supervised learning.
>
> We would also argue that MoCo, DINO, and VICReg are not all contrastive methods, as their loss function is not contrastive. As per [[3](https://arxiv.org/abs/2304.12210] we define DINO and VICReg as self-distillation and canonical correlation analysis approaches, respectively. While [[4](https://arxiv.org/abs/2206.02574)] (which we assume is the paper the reviewer is referring to) argue that VICReg and SimCLR can be made equivalent, the paper also directly states that other methods such as DINO, MoCo, and SimSiam do not perfectly fit this framework.
>
> > Writing and Presentation: I think the work can be improved a lot, with the current format.
>
> We are sorry to hear that the presentation was not satisfactory. We are more than happy to improve the writing and presentation of any part if the reviewer is willing to point these out to us.
>
> [1] A Comprehensive Survey on Deep Clustering: Taxonomy, Challenges, and Future Directions - Zhou et al. arXiv, 2022
>
> [2] DeepDPM: Deep Clustering With an Unknown Number of Clusters - Ronen et al. CVPR 2022
>
> [3] A Cookbook of Self-Supervised Learning - Balestriero et al. arXiv 2023
>
> [4] On the duality between contrastive and non-contrastive self-supervised learning - Garrido et al, ICLR 2022

---

### Official Review · Reviewer_bmQE · 2023-10-31

**Soundness:** 4 excellent
**Presentation:** 3 good
**Contribution:** 2 fair
**Rating:** 3
**Confidence:** 4

**Summary:**

- The paper considers the task of "zero-shot clustering" of feature embeddings from pretrained SSL networks, viz. MoCov3, DINO, VICReg, MAE, and CLIP
- It considers multiple clustering methods, K-means, Agglomerative clustering, Affinity Propagation, HDBSCAN and performs an extensive hyperparameter search on various hyperparameters
- The paper evaluates the performance of all these methods on each encoder for various datasets, like ImageNet-1K, CIFAR, iNaturalist-21, etc. and evaluates performance via AMI (Adjusted Mutual Information) which needs labels and Silhouette Scores, which doesn't require ground-truth labels
- The paper finds which encoders work the best with ResNet-50 and ViT-B architectures and which clustering methods perform the best -- MoCo-v3 for ResNet-50 and CLIP for ViT-B, with Agglomerative clustering performing the best
- Lastly, the paper shows that AMI and silhouette scores are well correlated, meaning that silhouette scores can be used as a good metric to evaluate how the clustering performs without having ground truth information

**Strengths:**

- The paper shows an extensive evaluation of clustering methods on various SSL methods and compares it to supervised and CLIP models
- It provides a good analysis of which methods are best for clustering, which clustering methods to use, and how to evaluate these even in the absence of ground truth labels
- The paper is well written and easy to read. The results are clear and easy to interpret.

**Weaknesses:**

- The paper is primarily a vast hyperparameter search of {pretrained models, clustering method, clustering hyperparams, eval datasets} and an analysis of the results. While the results are useful to use as a reference to pick the best models + clustering methods for a particular task, I fail to see any impactful research contributions. The analysis of correlation between AMI and Silhouette score is useful, but again not strong enough to warrant acceptance
- The results in Figure 1 and 2, especially Figure 1, have very wide error bars, meaning that the comparisons wouldn't be statistically significant. This is probably because of the use of the rank metric -- it would be better to see an aggregated performance metric which isn't as sensitive (like an average), resulting in some statistically significant results
- Minor: In the Introduction, the paper mentions "while the embeddings from an SSL-trained feature extractor can perform well on downstream tasks after fine-tuning the network, there has been little investigation into the utility of the embeddings without fine-tuning the network" -- this is inaccurate, SSL methods have been extensively evaluated without finetuning via k-nearest neighbours and linear probing while keeping the encoder frozen (e.g. MoCov3, DINO).
- Minor: the paper considers CLIP as a self-supervised method, whereas it is typically considered a weakly supervised method since the texts provide noisy supervision.

**Questions:**

- I am overall unsure what the rebuttal can show which can add research contributions to the paper. As of now, it seems like a study of clustering methods on SSL encoders. A study can also have interesting research observations / contributions but currently the paper seems like a simple description of the observations without any further analyses. Here's a few things which come to mind but I am still not sure if they'll be enough:
  - It would be interesting to see a connection between clustering scores and traditional SSL evaluations like k-NN, linear probes, or finetuned results -- do these move together in unison or are the clustering capabilities of a model's representations relatively unrelated to other evaluations?
  - Why does agglomerative clustering perform well, or what makes certain methods perform better? Why does changing the architecture change the rank ordering?

---

> ### Author Response · Authors · 2023-11-23
>
> We appreciate the time the reviewer has spent on analyzing and understanding the paper, and for highlighting the quality and extensiveness of the performed analysis.
>
> > The paper is primarily a vast hyperparameter search of {pretrained models, clustering method, clustering hyperparams, eval datasets} and an analysis of the results. While the results are useful to use as a reference to pick the best models + clustering methods for a particular task, I fail to see any impactful research contributions. The analysis of correlation between AMI and Silhouette score is useful, but again not strong enough to warrant acceptance
>
> We thank the reviewer for giving us the opportunity to highlight our finding that raw SSL features are not necessarily more clusterable than those from a supervised network is a novel insight, and worth diving deeper into in future research. This challenges the idea that SSL features should be more general than supervised features. We also want to clarify that we do not recommend any specific combination of encoder and clustering method. We instead recommend using a single clustering method—agglomerative clustering—for all tasks.
>
> > The results in Figure 1 and 2, especially Figure 1, have very wide error bars, meaning that the comparisons wouldn't be statistically significant. This is probably because of the use of the rank metric -- it would be better to see an aggregated performance metric which isn't as sensitive (like an average), resulting in some statistically significant results
>
> A rank test has fewer priors than a *t*-test and we don't think there is a reason to strongly assume that the delta in the accuracy between two approaches on different datasets would have a Gaussian distribution. Therefore, we are of the opinion that our current approach is better principled, but we would indeed probably get more statistical power by assuming the distance was comparable across datasets.
>
>
> > Minor: In the Introduction, the paper mentions "while the embeddings from an SSL-trained feature extractor can perform well on downstream tasks after fine-tuning the network, there has been little investigation into the utility of the embeddings without fine-tuning the network" -- this is inaccurate, SSL methods have been extensively evaluated without finetuning via k-nearest neighbours and linear probing while keeping the encoder frozen (e.g. MoCov3, DINO).
>
> We agree with the reviewers comment, and will correct the text to emphasise that the uniqueness to our approach comes not from the fact that there is no fine-tuning of the network, but that there is no training on labelled data from the new domain at all, and that the closest evaluation method would be kNN probing.
>
>
> > Minor: the paper considers CLIP as a self-supervised method, whereas it is typically considered a weakly supervised method since the texts provide noisy supervision.
>
> We agree that CLIP can be categorised as weakly supervised learning. However, as the model is learnt with a contrastive learning loss-function, we also argue that it is fair to categorise it as “cross-domain contrastive learning” and thereby as self-supervised learning.
>
> > It would be interesting to see a connection between clustering scores and traditional SSL evaluations like k-NN, linear probes, or finetuned results -- do these move together in unison or are the clustering capabilities of a model's representations relatively unrelated to other evaluations?
>
> This would be an interesting analysis that we hope to integrate in a future version of the manuscript.
>
> > As of now, it seems like a study of clustering methods on SSL encoders. A study can also have interesting research observations / contributions but currently the paper seems like a simple description of the observations without any further analyses
>
> These are all good questions. While we may provide some insights into why e.g. agglomerative clustering performs well by going deeper into a set of predictions, we will in general be hard-pressed to provide any guarantees. However, we would also argue that these exploratory studies are important as they highlight a hole in our general understanding of commonly used methods (in this case SSL pretraining), even if we cannot provide all the answers for why this is.

---

### Official Review · Reviewer_XtjS · 2023-11-01

**Soundness:** 2 fair
**Presentation:** 2 fair
**Contribution:** 3 good
**Rating:** 5
**Confidence:** 4

**Summary:**

The paper presents an empirical study on zero-shot clustering utilizing SSL pre-trained models. The study encompasses analysis on 10 representative datasets, focusing on 5 prevalent pre-training paradigms (MoCo, DINO, VICReg, MAE, and CLIP) with ReNet and ViT backbones. The findings reveal insightful tendencies, such as the potential of contrastive and multi-modal SSL models in generating meaningful clusters. The paper is interesting in general, but some major issues should be addressed.

**Strengths:**

1. The paper serves as the pioneering investigation into zero-shot clustering of SSL feature encoders.
2. The experimental design is comprehensive, encompassing diverse datasets, pre-trained models, and clustering methods.
3. The experimental results yield valuable insights into zero-shot clustering, providing valuable guidance for future model development.

**Weaknesses:**

1. The paper addresses the concept of zero-shot clustering, an important subject in machine learning. However, there is a need for a clearer definition of the zero-shot clustering problem. Additionally, the paper should thoroughly review and discuss related topics, such as transfer learning or unsupervised domain adaptation, for a comprehensive understanding.
2. In Section 3, further elaboration is required on the training process of the supervised model. It would be beneficial to include a table outlining different SSL models and their respective configurations for enhanced clarity.
3. In order to ensure a fair comparison, the paper should consider comparing deep models initialized with random weights, as in many cases, random mapping can yield clustering results that surpass those of raw data.
4. While the paper highlights the improved performance of clustering methods with pre-trained models, it lacks an exploration of the underlying reasons for this phenomenon.
5. Visual comparisons, such as t-SNE plots, are essential in the experiment to provide a more comprehensive evaluation.
6. The paper mentioned dimension reduction in the experiment but did not discuss the effect of them.

**Questions:**

please see the weaknesses

---

> ### Author Response · Authors · 2023-11-23
>
> We appreciate the thoughtful review and the recognition of the comprehensiveness of our study and the contained results.
>
> > The paper addresses the concept of zero-shot clustering, an important subject in machine learning. However, there is a need for a clearer definition of the zero-shot clustering problem
>
> We agree that the omission of a clear definition of “zero-shot clustering” is unfortunate and we will make sure that this will be included in a future version.
>
> > The paper should thoroughly review and discuss related topics, such as transfer learning or unsupervised domain adaptation, for a comprehensive understanding
>
> We agree that the mentioned fields are highly relevant and the literature review should be extended to include these fields.
>
> > Further elaboration is required on the training process of the supervised model.
>
> We used the pretrained models available in TorchVision v0.16, [[ResNet50](https://pytorch.org/vision/stable/models/generated/torchvision.models.resnet50.html#torchvision.models.ResNet50_Weights )] [[ViT-B](https://pytorch.org/vision/stable/models/generated/torchvision.models.vit_b_16.html#torchvision.models.ViT_B_16_Weights )]. We will make sure that this is clearly stated in the manuscript.
>
> > It would be beneficial to include a table outlining different SSL models and their respective configurations for enhanced clarity.
>
> Thank you for the suggestion. We will add a summary table indicating the training configurations of each model to the supplementary materials.
>
> > In order to ensure a fair comparison, the paper should consider comparing deep models initialized with random weights, as in many cases, random mapping can yield clustering results that surpass those of raw data.
>
> This is an excellent idea that we appreciate the reviewer for highlighting. In a future version of the manuscript we will include performance using randomly initialized weights as a baseline.
>
> > While the paper highlights the improved performance of clustering methods with pre-trained models, it lacks an exploration of the underlying reasons for this phenomenon.
>
> In general SSL-pretraining has been found to perform well for downstream tasks such as classification, segmentation, and detection. This was the motivation for investigating how well the features can be clustered, without fine-tuning any part of the network.
>
> > Visual comparisons, such as t-SNE plots, are essential in the experiment to provide a more comprehensive evaluation.
>
> Thank you for the suggestion. We will provide visual examples of clusters in a future version of the supplementary materials, in order to give the reader a better understanding of what is happening.
>
> > The paper mentioned dimension reduction in the experiment but did not discuss the effect of them
>
> The effect of dimensionality reduction is partially covered in section 3.5, covering the hyperparameter search. However, we are interested in further quantifying the effect of dimensionality reduction in a future version of the manuscript.

---

### Meta-Review · Area_Chair_DPAK · 2023-12-07

**Metareview:**

This paper considers an approach to the zero-shot clustering problem.

The reviewers were generally in agreement that this paper is not yet ready for publication, with all four reviewers advocating for rejecting the paper.  They noted several issues, including clarity of the paper (e.g., a clear a definition of the problem studied), a need for additional experiments, and limited research contributions of the approach.  The authors did provide a rebuttal, which did address some of these concerns, but reviewers maintained their scores after the rebuttal period.  Thus, it seems that the paper could use additional work before it is ready for publication.  Please keep in mind the suggestions of the reviewers when preparing a future version of the manuscript.

**Justification For Why Not Higher Score:**

All reviewers are in agreement here, and the rebuttal did not sway them.  There are some fundamental concerns that were not addressed by the rebuttal.

**Justification For Why Not Lower Score:**

N/A

---

### Decision · Program_Chairs · 2024-01-16

Reject